# Hybrid Decision-Making-Method-Based Intelligent System for Integrated Bogie Welding Manufacturing

**Kainan Guan** [1,2]**, Yang Sun** [1,2]**, Guang Yang** [1,2] **and Xinhua Yang** [1,2,*]

1     School of Materials Science and Engineering, Dalian Jiaotong University, Dalian 116028, China
2     Liaoning Key Laboratory of Welding and Reliability of Rail Transportation Equipment,
      Dalian Jiaotong University, Dalian 116028, China
*     Correspondence: yangxh@djtu.edu.cn

**Featured Application: Parts of our research have been applied to decision making in the welding manufacturing process of rail vehicles. We predict that this research will be of value to the design of welding manufacturing process decisions in different industries.**

**Abstract:** To address the challenges of incomplete knowledge representation, independent decision ranges, and insufficient causal decisions in bogie welding decisions, this paper proposes a hybrid decision-making method and develops a corresponding intelligent system. The collaborative case, rule, and knowledge graph approach is used to support structured documents and domain causality decisions. In addition, we created a knowledge model of bogie welding characteristics and proposed a case-matching method based on empirical weights. Several entity categorizations and relationship extraction models were trained under supervised conditions while building the knowledge graph. CRF and CR-CNN obtained high combined F1 scores (0.710 for CRF and 0.802 for CR-CNN) in the entity classification and relationship extraction tasks, respectively. We designed and developed an intelligent decision system based on the proposed method to implement engineering applications. This system was validated with some actual engineering data. The results show that the system obtained a high score on the accuracy test (0.947 for Corrected Accuracy) and can effectively complete structured document and causality decision-making tasks, having large research significance and engineering value.

**Keywords:** welding; case-based; rule-based; knowledge graph; entity classification; relationship extraction





## 1. Introduction

The bogie is one of the major components of the rail vehicle, which is mainly manufactured via welding. The welding process is a multi-stage collaborative process that generally includes structural design, process development, production, and quality control. The accuracy and synergy of the decision making in all phases directly determine the productivity, safety, and comfort of rail vehicles. The traditional operation of the welding process primarily relies on the decisions of domain experts. This makes the workload high, the standardization low, and the errors frequent because of the arbitrariness and uncertainty of manual operation. Knowledge-based intelligent systems [1–6] have received great attention for manual decision-making challenges. In these systems, expert knowledge is translated into computer language to assist in designing the welding manufacturing process, and welding process systems [7] and fault diagnosis systems [8] have been widely applied in welding engineering. Digital-based engineering decisions have become an essential direction for upgrading manufacturing models. However, due to the high-quality requirements, standards, and engineering complexity of bogie welding, the manufacturing process is still dominated by manual decisions, supplemented only by intelligent decisions in single-stage manufacturing. Thus, introducing intelligent decisions into integrated welding manufacturing would have significant engineering and social value.

In recent years, decision reasoning methods have mainly focused on case-based [9,10], rule-based [11,12], and hybrid decision methods [13–15]. Case-based decision making is the process of obtaining the best matching case by calculating the similarity between old and new cases. This technique can be used to make decisions about future events based on a complete experience database, so it is widely employed in several domains, such as assembly management [16], process engineering design [17], and fault diagnosis [18]. However, effective decision making may be challenging when there is a lack of a case base. Rule-based methods [19] are also often used for decision making, where a priori experience and knowledge are transformed into mathematical logic to carry out decision tasks. Compared with the case-based approach, the rule-based system reduces the reliance on a priori data, but complex logic construction still needs to be solved. In order to synergize the advantages of case-based and rule-based approaches, hybrid decision-making methods have become a major research focus. Phyu et al. [20] proposed a hybrid approach to rule- and case-based material failure analysis and demonstrated that the hybrid approach obtains better analytical results than individual decisions. In weld manufacturing, Zhang et al. [21] offered a hybrid case- and rule-based system for welding processes and implemented system-level applications in weld manufacturing. In addition, algorithms such as Bayesian [22], data envelopment analysis [23], multi-criteria [24–26], fuzzy theory [27–30], and neural networks [31,32] have been introduced into decision making. These approaches have shown positive performance in several domains. However, incomplete knowledge representations are not addressed further due to data and logical form limitations. Secondly, most welding systems focus on a single stage of the production cycle and need to be adapted to adequately meet the integrated manufacturing model. Thirdly, end-to-end decision making alone does not satisfy the search for causality.

A hybrid decision method based on a knowledge graph is proposed for the challenges of incomplete knowledge representation, independent decision scope, and insufficient causal decisions in integrated bogie welding manufacturing. This approach is centered on a knowledge graph and collaborates with rule-based and case-based decision methods to enable full-cycle decision making for welding and to support causal decision making across production stages. Compared with other methods, the hybrid decision method has the following characteristics: (i) Wide range of decision making: this method can make decisions not only for structured documents but also for unstructured causal relationships. (ii) Diverse knowledge expressions: structured data and knowledge can be characterized based on cases or rules, and unstructured semantic information is also characterized via the construction of domain knowledge graphs. (iii) High engineering applicability: the decision process involves the entire lifecycle of welding manufacturing, providing decision results with cause-and-effect relationships that apply to all phases of the guiding decision-making process.

In this study, a hybrid decision system for bogie welding manufacturing was developed based on the proposed method. The decision system is divided into two subtasks: structured document decision making and causal decision making. Among other things, structured documents such as welding process specifications (WPS), weld joint lists, and welding plans are used to make decisions based on rules and cases. A case database is constructed through case representation, case retrieval, and case revision. We store the data in a structured relational database to characterize the cases and propose a case-matching method based on empirical weights. Rule-based decision making is used to modify cases that have no matching prior cases or need to be corrected. To accomplish causal decision making, we created a top-down domain knowledge model based on the production process and created a corresponding knowledge graph. Knowledge graph creation consists of two subtasks: entity recognition and relationship extraction, and several classical models are trained based on supervised learning. The top-performing models are conditional random field (CRF) and classifying relations by ranking with convolutional neural networks (CR-CNN). They are used for entity recognition and relationship extraction to achieve unstructured automatic extraction, respectively. System functionality through modular design. Data design and storage are carried out using Mysql and Neo4j. The implementa-

tion results and user interface are presented in the results section. The results show that the intelligent decision system based on the proposed method can effectively complete the decision-making tasks for the whole life cycle of bogie welding, which has high research significance and engineering value. The main contributions of this paper are as follows:

(i).    We innovatively joined knowledge graph decision making to case- and rule-based methods and applied this hybrid model to integrated bogie welding manufacturing.

(ii).   A production process-based knowledge model is developed to support knowledge system design. An empirical weight-based approach is proposed to calculate the case similarity.

(iii).  We developed an intelligent decision system based on a hybrid decision model in conjunction with welding manufacturing, which can complete structured documents and cause-and-effect decisions.

The rest of this paper is organized as follows. In Section 2, the knowledge model and hybrid decision scheme are constructed for the bogie welding manufacturing process. Section 3 implements the hybrid decision scheme based on case matching, rule base, and knowledge graph construction. Section 4 gives the resultant metrics of the comparative models for entity identification and relationship extraction in the construction of domain knowledge graphs. In addition, experimental results on the knowledge graph construction process, structured data decision making, and causal decision making are provided and analyzed. In Section 5, the corresponding conclusions are given.

## 2. Methods and Models

### 2.1. The Bogie Manufacturing Process

As a critical component of rail vehicles, the bogie improves load, vehicle guidance, and cushioning damping. Its main components are a rocker, side beam, frame, axle box, spring damping, braking device, etc. The composition of the bogie's structure is shown in Figure 1. The bogie manufacturing process includes many welding processes, and implementing intelligent welding decisions benefits productivity and improves quality. Welding manufacturing is a complex process of multi-departmental collaboration, mainly including structural design, process selection, manufacturing, quality inspection, and experimentation. The structural design phase is mainly completed by determining geometric structure information such as static dimensions, fatigue dimensions, and joint forms. Next, in the process selection stage, a suitable process is selected according to the structural information and relevant standards and then issued to production. Structural and process information will be considered in the production phase to complete the assembly and welding of the parts. Molded parts that meet the quality inspection requirements are used for vehicle assembly. In addition, parts with quality problems need to be traced back to the cause and repaired to ensure manufacturing quality.

As described above, the bogie welding process involves multi-department, multi-component, and multi-station information, and its manufacturing process requires a great deal of decision making. Expert experience and manual decisions alone do not easily guarantee manufacturing quality and efficiency. Therefore, replacing manual decision making with computers will help reduce manual casualness, avoid repetitive work, and improve decision making efficiency.

### 2.2. Domain Knowledge Modeling

Knowledge modeling is the process of structurally summarizing and characterizing domain knowledge. Proper structural design can effectively reduce knowledge redundancy and improve application efficiency. The data and application characteristics are essential factors in designing the model structure. The bogie welding manufacturing process includes several stages of structural design, process selection, production operations, and quality inspection, which have evident process characteristics. In addition, each assignment phase corresponds to some specific professional content. For example, the structural design includes joint design and quality level. Process selection includes welding method and

position selection. Therefore, based on the domain data characteristics, we proposed a collaborative production process and top-down fusion approach to achieve model structure design. The operation phase is the first level of classification information, and its sub-information is classified sequentially based on a top-down approach. The domain knowledge model is shown in Figure 2.

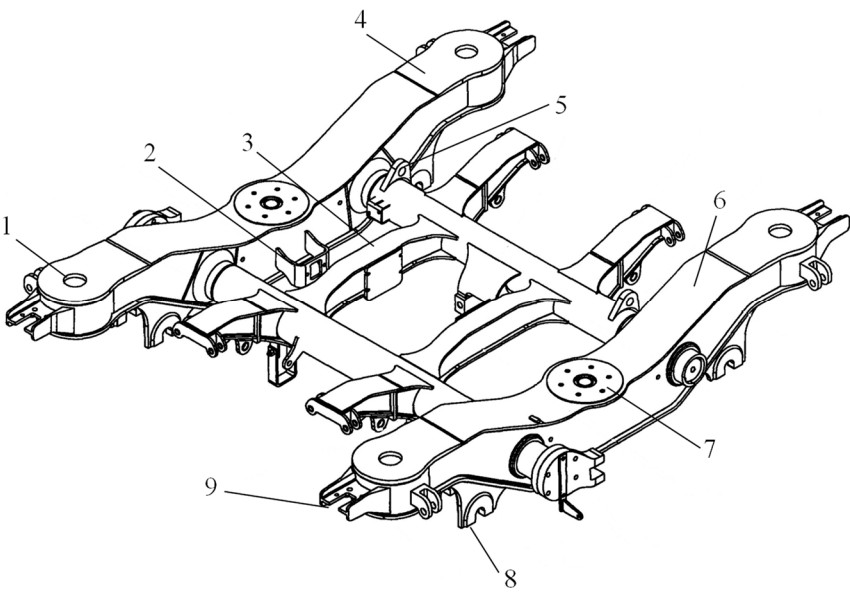

**Figure 1.** Description of bogie structure. 1—Framework components; 2—Series II transverse shock absorbers seats; 3—Crossbeam components; 4—Spring mounts; 5—Lifting seat; 6—Side beam components; 7—Air spring seats; 8—Axle box locator; 9—Series I pendant locators.

| Welding | | | | |
|---|---|---|---|---|
| Standards | Design | Technology | Manufacture | Quality |
| International Standards | Static size | Electric current | Workstation | Experiment |
| National Standards | Fatigue size | Voltage | Sanding | Visual inspection |
| Enterprise standards | Stress level | Welding speed | Flattening | Magnetic particle |
| Regulations | Defect level | Welding position | Bending | Defects |
| Production requirements | Quality grade | Welding method | Painting | Residual stress |
| Indicator requirements | Joint | Preheat temperature | Components | Welding distortion |
| …… | …… | …… | …… | …… |
| Department | | | | |

**Figure 2.** The welding knowledge model.

The model structure is also designed considering the association between nodes and the type of application. The purpose of the model is to summarize the entities involved in the bogie welding process and to support decision making via cause-and-effect relationships between entities. Thus, decision problems are divided into cause-based, result-based, and other decisions, for example, Q1: "what attributes support the decision of attribute $A$?"; Q2: "which attribute is selected based on attributes $A$ and $B$?"; Q3: "what attributes do attribute $C$ contain?". The tracing of reasons for decision attributes is called the cause-based decision,

as in the Q1 class of queries. The process of determining the results' attributes from known attributes is a results-based decision, as described in the example in Q2. In addition, decisions on attribute ontology features, such as in Q3, are classified as other decisions.

### 2.3. Collaborative Decision Making

Depending on the target task, the welding manufacturing decision process can be divided into directed and undirected decisions. Directed decisions are mainly aimed at standardized documents, such as welding process specifications (WPS), welding plans, and inspection plans. Decision attributes are largely fixed and rely on a large amount of experience and structured data. Undirected decisions are used to solve unstructured semantic problems with random attributes. Considering the characteristics of welding manufacturing data, we propose a collaborative decision-making scheme that employs case-based reasoning (CBR) and rule-based reasoning (RBR) for attribute-specific decisions and constructs knowledge graphs for undirected decisions. The overall process is shown in Figure 3.

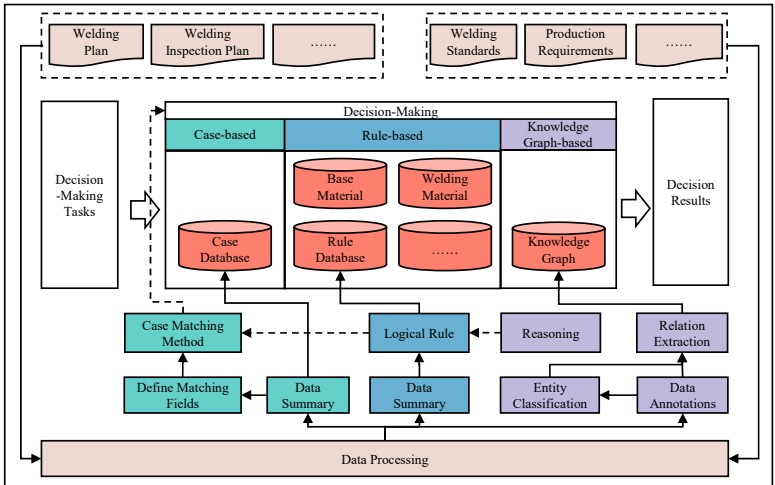

**Figure 3.** Hybrid decision-making process.

(1) Case-based decision making. The CBR is a method for completing decisions by analyzing the similarity between a problem and a known case base. Its main elements are case representation, case retrieval, case revision, and case learning. Case representation is the process of designing the rational storage form of data, and the characteristics of actual cases need to be considered. In this study, we divide the data into conditional and result fields and store them in a structured database. Case retrieval is a crucial step of CBR, which realizes case matching by calculating the similarity between cases. When the similarity of the condition fields is greater than a certain threshold for a decision problem, we consider the result field a reasonable decision. Case revision mainly consists of modifying invalid decisions into reasonable decisions. In order to continuously expand the case base and make it have a strong generalization ability, it is necessary to supplement the revised reasonable cases to the case base. This process is called case learning. The decision process is shown in Figure 4.

(2) Rule-based decision making. The RBR is the process of transforming domain expert knowledge into computer logic language to complete decision making. The main steps to complete the decision are the representation, acquisition, and reasoning of knowledge. From the perspective of knowledge, decision making is a process from knowledge representation to knowledge acquisition. Knowledge is generally divided into factual knowledge and process knowledge. Factual knowledge is the basic description of things and represents the characteristics of individual attributes. For example, we can immediately understand the welding angle when referring to

the welding position. Process knowledge is a collection of knowledge obtained by some means or logical operation. For example, the preheating temperature needs to compare to the preheating temperature of base-metal-1 and base-metal-2 and select the minimum temperature value. According to different sets of knowledge, choosing different knowledge expression methods is the key to realizing process reasoning. Rules are the logical expression of knowledge relations and expert experience. Rules can be divided into dynamic and logical rules according to different action modes. Dynamic rules refer to the correspondence between fields. They define fields A, B, and C. When A and B occur, C must occur. Therefore, there is a correspondence between A, B, and C. This correspondence is called a dynamic rule. Logical rules refer to the description statements connected with logical expressions, such as defining fields A, B, C, and D, and there is a logical statement "IF A > B THEN C, ELSE D," which is the logical rule between fields.

(3) Knowledge graph-based decision making. Knowledge graph-based decision-making transforms unstructured natural language into structured query statements and retrieves the graph for the best matching answer. Identifying keywords and relationships is a vital part of the search and decision making. The automatic relationship extraction model is trained and used to define interrogative relationship categories in the knowledge graph's construction. In addition, the trained entity recognition model is also used as a preliminary determination of keywords. Next, the Term Frequency-Inverse Document Frequency (TF-IDF) method is employed to identify the most critical terms. The TF-IDF is calculated according to Equations (1)–(3), and the larger its value, the more critical the corresponding vocabulary. Structured statements are created based on keywords and relationships and are used to retrieve decision results.

$$\mathrm{TF} - \mathrm{IDF} = \mathrm{TF}_{ij} \times \mathrm{IDF}_i \tag{1}$$

$$\mathrm{TF}_{ij} = \frac{n_{ij}}{\sum_k n_{kj}} \tag{2}$$

$$\mathrm{IDF}_i = \log\left(\frac{|D|}{1 + |\{j : t_i \in d_j\}|}\right) \tag{3}$$

where $n_{ij}$ is the number of occurrences of the target lexical entry in the semantics, and the summation $n_{kj}$ represents the number of all lexical entries. $D$ is the total number of lexical entities in the corpus. $\{j{:}t_i \in d_j\}$ is the number of corpora containing $t_i$.

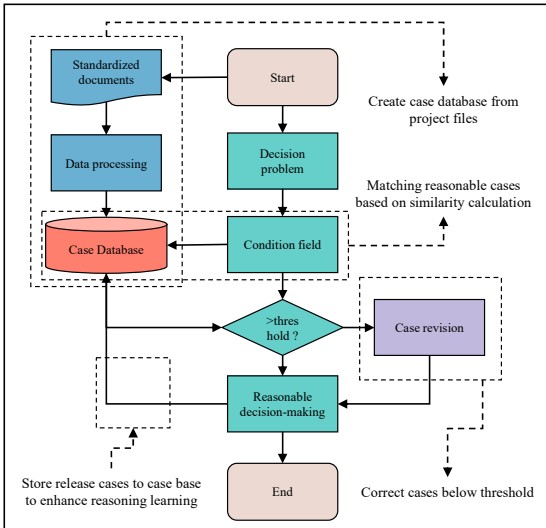

**Figure 4.** CBR-based decision-making process.

## 3. Experiments

### 3.1. The System Design

A function-based modular design is adopted to reduce the coupling between functional modules. System functions are divided into five modules: system management, graph creation, graph list, business decision, and business approval. System management realizes authority division by assigning corresponding function modules to users. Graph creation and lists are used for knowledge graph creation and presentation, and the creative process includes manual and automatic creation. The business decision module realizes the decision of structured files and causality based on the constructed knowledge system. In addition, the decision results processed and issued via the approval module will be considered formal decisions.

### 3.2. Build Cases and Rules

Case construction is divided into three subtasks: case representation, case retrieval, and case revision. Case representation can also be seen as a form of case storage. Depending on the data characteristics of the weld file, cases are stored in a structured database. Case retrieval is a critical element of the acquisition decision, and the best matching result is obtained by calculating the similarity between new and old cases. The similarity calculation compares the problem with the known cases to obtain the highest matching result to complete the decision. In order to obtain credible matching results, the computational process must consider the attribute characteristics of the target data. As far as welding decisions are concerned, each condition attribute has a different degree of influence on the result attribute. It is necessary to define the weights for each attribute. The steps of similarity calculation based on weight are as follows:

(1)  Case similarity calculation. The new case is defined as P, the old case as C, and the similarity calculation process between cases is shown in Equation (4). We calculate the product of the similarity of each attribute and its weight and sum it to obtain the case similarity. Here, sim (P, C) represents the case similarity, n is the number of case condition attributes, ai represents the i-th attribute, and $\omega_i$ represents the i-th attribute weight:

$$\text{sim}(P, C) = \sum_{i=1}^{n} \omega_i \cdot \text{sim}\left(a_i^P, a_i^C\right) \tag{4}$$

(2)  Attribute similarity calculation. Attributes generally include numeric, symbolic, and conforming attributes. Equation (5) is the numerical attribute calculation process. For symbolic attributes, the attribute similarity is "1" when the attribute values are equal and "0" when the attribute values are unequal. The calculation process is shown in Equation (6). Composite attributes contain numeric and symbolic characteristics. Among them, the symbolic feature is selected as the first matching element. The similarity is calculated according to Equation (5), when the symbols are equal. If the symbols are not equal, the similarity is "0":

$$\text{sim}\left(a_i^P, a_i^C\right) = \frac{1}{m^{|a_i^P - a_i^C|}}, \tag{5}$$

$$\text{sim}\left(a_i^P, a_i^C\right) = \begin{cases} 1, a_i^P = a_i^C \\ 0, a_i^P \neq a_i^C \end{cases}, \tag{6}$$

where $a_i^P$ is the i-th attribute value of the new case, $a_i^C$ is the i-th attribute value of the old case, and m is constant (*m* = 1.2 according to field experience).

(3)  Attribute weight division. Assigning the same weight to the attributes is unreasonable because the conditional attributes in the case have different degrees of influence on the decision attributes. Therefore, the conditional attribute weights are calculated

based on the number of decision attributes associated with the attribute, which is calculated as Equation (7), where ai denotes the i-th conditional attribute and count(ai) denotes the number of decision attributes associated with the conditional attribute:

$$\omega_i = \frac{\text{count}(a_i)}{\sum\limits_{i=1}^{n} \text{count}(a_i)} \tag{7}$$

Based on expert experience, 0.85 is chosen as the case-matching threshold, and cases below the threshold do not have reusability. For valid cases, case revision is required to ensure the usability of the case. Rule-based decision-making methods are employed to revise the case. Rule construction is the process of transforming expert experience into an executable computer program. Logic rules are divided into dynamic rules and static rules. Dynamic rules rely on database calls to build the logic. For example, the decision rules for welding procedure qualification records (WPQR) require a structural database correspondence to complete the development. Rules constructed only through logical relations and predicates are called static rules, for example, "IF weld quality level == CP A THEN weld inspection level == CT 1". In this study, 20 rules were constructed involving several required fields such as welding position, preheating temperature, quality level, and welder qualification. Constructed rules are used as case corrections, and approved cases for distribution are added to the case base to improve data quality.

*3.3. Building a Knowledge Graph*

Creating a knowledge graph is a key element in enabling decision making. Entities and relationships are the essential elements of knowledge, so fast extraction of entities and relationships from unstructured data is necessary. An automatic method of data extraction based on manual creation was designed in this study. The process of extraction and creation is as follows:

(1) Data processing. The purpose of the design knowledge graph is to provide guided decisions for the bogie welding manufacturing process. Domain guidance documents are selected, such as domain standards, regulations, and production requirements. Most of these data are unstructured files and cannot be extracted directly. In order to obtain valid research data, the document information is split into several sentences based on essential separators such as full stops, exclamation marks, question marks, and semicolons. Sentence-level data are saved as a data source and divided into several attribute vocabularies via the CRF model [33]. The vocabulary data and terminology were converted into numerical vectors with the skip-gram model of Word2vec. Sentence-level, word-level, and word vector data are used to support the extraction of entities and relationships. In addition, the Begin-Inside-Outside (BIO) approach is employed to label the entity vocabulary dataset for entity identification. Labels and sentence-level semantics are defined to support relation extraction.

(2) Entity extraction. Based on the established knowledge model, semantic relationships are classified into six categories: "Design," "Technology," "Manufacture," "Quality," "Department," and "Standard." Subordinate production process attributes have a small amount of data with many categories, so high-quality models are challenging to train. The relation "belong_to" is used as a link between the production process category and the subordinate attributes to replace the classification of the subordinate entities. This relation is also trained in the relation extraction task. Several models based on supervised learning Hidden Markov Model (HMM), CRF, and Bi-directional Long Short-Term Memory (BiLSTM) were trained on the same dataset. Higher-quality models are employed to extract domain entities automatically.

(3) Relationship extraction. Relationships are the ties linking entities and are important supporting information for knowledge graph decisions. This study divided the relationships into five main categories: belong_to, reference, requirement, applicable_to, and unknown. The "belong_to" category is used to associate upper-level and

lower-level category attributes. We use relation "reference" to express the retroactive relationship in decision making. Equally, relations "requirement" and "applicable_to" are defined to support cause-based decision making. The Bi-directional Long Short-Term Memory and Attention (BiLSTM + Attention) [34] and CR-CNN [35] models for relation extraction are trained under supervision.

*3.4. Database Design*

A fusion decision-making method centered on knowledge graphs, collaborative CBR, and RBR is proposed for bogie welding manufacturing. Its functional implementation requires the support of a graph database and a structured relational database. The relational database includes a terminology dictionary, base materials, welding procedure qualification records, welder qualifications, welding materials, and a welding parameters database. An MYSQL database was employed and built for structured document decisions and partial business execution. Graph databases are mainly used to support the construction of knowledge graphs because of their compatibility with complex relationships. Neo4j was chosen as the graph data creation tool, which stores node, relationship, attribute, and label information in arrays.

## 4. Results and Interpretability

*4.1. Model Training Results*

Metrics are essential to evaluating the quality of a model. We combined the influence of the number of data categories and selected Precision, Recall, and F1-score as model evaluation metrics. The calculation process and variable information are shown in Table 1.

**Table 1.** Description of model validation metrics.

| No. | Equations | Variable Interpretation |
|---|---|---|
| 1 | $\text{Precision} = \dfrac{\text{TP}}{\text{TP} + \text{FP}}$ | TP (True Positive): both true and predicted categories are positive examples. |
| | | FP (False Positive): true category is negative and predicted category is positive. |
| 2 | $\text{Recall} = \dfrac{\text{TP}}{\text{TP} + \text{FN}}$ | TN (True Negative): both true and predicted categories are negative examples. |
| | | FN (False Negative): true category is positive and predicted category is negative. |
| 3 | $\text{F1-score} = \dfrac{2 \times \text{Precision} \times \text{Recall}}{\text{Precision} + \text{Recall}}$ | Precision and Recall refer to the equations numbered 1 and 2, respectively. |
| | | F1-score is calculated by the equation numbered 3. |

For the dataset, 19,410 sample pairs of entity-labeled data were obtained and divided into a training set, a test set, and a validation set. We manually checked and adjusted some data to ensure that the partitioned data were evenly distributed in each category. The training set contains 15,113 pairs of data samples, the test set involves 2350 pairs of samples, and the validation set has 1947 pairs of samples. In addition, to compare the training relationship extraction models, we divided the 1832 pairs of labeled sentence-level data into a training set, a test set, and a validation set in a ratio of 8:1:1. The entity and relationship extraction training results based on Precision, Recall and F1-score metrics are presented in Tables 2 and 3, respectively.

As the results show, CRF obtained a high score in the entity extraction model, which was employed to complete the automatic entity extraction task. For the relationship extraction task, CR-CNN model achieved a higher F score for "belong_to," "reference," and "requirement." The BiLSTM + Attention model performed better in the other categories and had a higher overall score. Since the category "unknown" positively contributes to the composite score of BiLSTM + Attention, it has low importance in the relationship category. Therefore, we combine the other category F1-scores and employ CR-CNN to implement the relationship auto-extraction task. In addition, we consider that annotation quality and feature acquisition methods may be important factors for improving model quality.

**Table 2.** Entity extraction model results metrics.

| Model | Metrics | Technology | Manufacture | Design | Department | Standard | Quality |
|---|---|---|---|---|---|---|---|
| HMM | Precision | 0.512 | 0.569 | 0.593 | 0.556 | 0.905 | 0.713 |
| | Recall | 0.528 | 0.474 | 0.743 | 0.590 | 0.880 | 0.599 |
| | F1-score | 0.519 | 0.517 | 0.646 | 0.569 | 0.892 | 0.650 |
| CRF | Precision | 0.783 | 0.686 | 0.715 | 0.722 | 0.935 | 0.758 |
| | Recall | 0.603 | 0.429 | 0.788 | 0.501 | 0.891 | 0.671 |
| | F1-score | 0.678 | 0.527 | 0.748 | 0.586 | 0.912 | 0.710 |
| BiLSTM | Precision | 0.613 | 0.476 | 0.592 | 0.662 | 0.937 | 0.569 |
| | Recall | 0.386 | 0.463 | 0.542 | 0.497 | 0.818 | 0.497 |
| | F1-score | 0.473 | 0.469 | 0.535 | 0.568 | 0.873 | 0.530 |

**Table 3.** Relationship extraction model results metrics.

| Models | Metrics | Belong_to | Reference | Requirement | Applicable_to | Unknown | Macro-Average |
|---|---|---|---|---|---|---|---|
| BiLSTM + Attention | Precision | 0.838 | 0.568 | 0.806 | 0.857 | 0.872 | 0.788 |
| | Recall | 1.000 | 0.875 | 0.625 | 0.878 | 0.850 | 0.846 |
| | F1-score | 0.912 | 0.688 | 0.704 | 0.867 | 0.861 | 0.816 |
| CR-CNN | Precision | 0.912 | 0.704 | 0.670 | 0.818 | 0.850 | 0.791 |
| | Recall | 1.000 | 0.792 | 0.763 | 0.878 | 0.638 | 0.814 |
| | F1-score | 0.954 | 0.745 | 0.713 | 0.847 | 0.729 | 0.802 |

### 4.2. Intelligent Decision Results

The 284 welding process data for bogie welding were used for system testing, including 107 side beams, 55 cross beams, 55 frames, and 22 brake hangers and beams. The data are mainly derived from WPS data that have been approved by experts and applied in engineering production. We defined design information as known data (type of base material, size of base material, quality grade, joint form, etc.) and process information (welding position, assembly gap, blunt edge, welding parameters, preheating temperature, etc.) as decision data. The model completes validates the decision information based on known information. The ratio of correct cases to the number of test cases, called Case Accuracy, is used to evaluate the accuracy of case matching. The accuracy of corrected cases (Corrected Accuracy) is calculated by the ratio of correct cases (Case Accuracy) to test cases (Test Cases). The test results are shown in Table 4.

**Table 4.** Structured document decision accuracy.

| No. | Part Name | Test Cases | Right Case | Case Accuracy | Corrected Case | Corrected Accuracy |
|---|---|---|---|---|---|---|
| 1 | Side Beam | 107 | 90 | 0.841 | 103 | 0.963 |
| 2 | Vehicle Frame | 110 | 92 | 0.836 | 107 | 0.973 |
| 3 | Crossbeam | 55 | 35 | 0.636 | 49 | 0.891 |
| 4 | Brake hanger/beam | 12 | 8 | 0.667 | 10 | 0.833 |
| Total | - | 284 | 225 | 0.792 | 269 | 0.947 |

As shown in Table 4, the case accuracy is 0.792, and the corrected accuracy is 0.947. Compared with the case-matching results, the corrected case significantly improved accuracy. The low case matching accuracy is mainly due to the incomplete case base, especially the lack of "Crossbeam" and "Brake hanger/beam" cases. In addition, the correctness of the input known conditions also affects the matching accuracy. Even though the corrected case achieves high accuracy, the limitation that the logical representation is designed for most problems and those individual problems are easily ignored has a negative impact on accuracy. Therefore, the decision accuracy of the system may be improved by enhancing the iterative ability of the case base, monitoring the condition input, and improving the rule compatibility.

For causal decision making, 10 decision questions were selected for testing the system. These issues involve multiple relationships ("belong_to," "reference," "requirement," applicable_to) and categories ("Design," "Technology," "Manufacture," "Quality," "Department," "Standard"). The decision problems and results are presented in Table 5 as unstructured natural language. As the results show, nine questions obtained corresponding decision results that are consistent with the relevant standards, and experience and can provide effective guidance for the bogie welding manufacturing process. However, obtaining a valid result is challenging when multiple entities or relationships are involved in the decision. Therefore, how to optimize the relationship and entity extraction strategy for the welding manufacturing process and design a relational storage structure for complex problem solving may become our next research focus.

**Table 5.** Causal decision results.

| No. | Decision Issues | Decision Results |
|---|---|---|
| 1 | What check grade is required for weld grade CP A? | The weld grade CP A requires check grade CT 1 |
| 2 | How to choose the base material for welding. | The base material reference CEN ISO/TR 15,608, weld grade |
| 3 | What properties determine the preheating temperature? | The preheating temperature reference material, plate thickness |
| 4 | How to determine the welding position. | The welding position reference joint and weld seam form |
| 5 | What does the welding position include? | The welding position includes PA, PB, PC, PD, PE |
| 6 | What welding position is required for a-joint? | The a-joint requires welding position PB |
| 7 | What assembly gap is selected for welding method t135 and plate thickness of 14? | No Result |
| 8 | Which department designs the welded joints? | The design department tasks include welded joint design |
| 9 | What parts are carbon steel applied to? | Carbon steel is applied to side beam, vehicle frame, crossbeam |
| 10 | What does Standard EN ISO 6520 apply to? | The Standard EN ISO 6520 is applied to defect classification |

*4.3. System Realization*

Hybrid decision-making methods are applied to the bogie welding process in the form of intelligent systems, which consist mainly of structured documented decisions and cause-and-effect-based, guided decision-making tasks. Structured documented decisions are used for the rapid development of WPS, welding schedules, inspection schedules, etc. The information to be decided is obtained by pre-inputting known data and based on a decision inference system to guide the welding production. Cause-and-effect-based decision making is mainly used for unstructured data. For example, the rapid determination of relevant standards in joint design, the acquisition of recommendations for inspection levels and means during quality inspection, and the provision of reference information for the traceability of welding defects.

The system mainly applies to welding decision making for structure documentation and causal issues. Vue is employed to implement page design, and JAVA is used for system business implementation. Structured file decisions are shown in Figures 5 and 6. The system enables importing conditional data through single additions and external Excel files, and the data can be modified through edit and delete buttons. Imported data are batch decided by selecting conditional data and clicking on the decision button. The structured file after the decision is presented as shown in Figure 6, which includes information such as condition data, welding parameters, and joint images. These documents are approved to guide the welding manufacturing process directly and are stored in a case database for optimization and iteration of case retrieval.

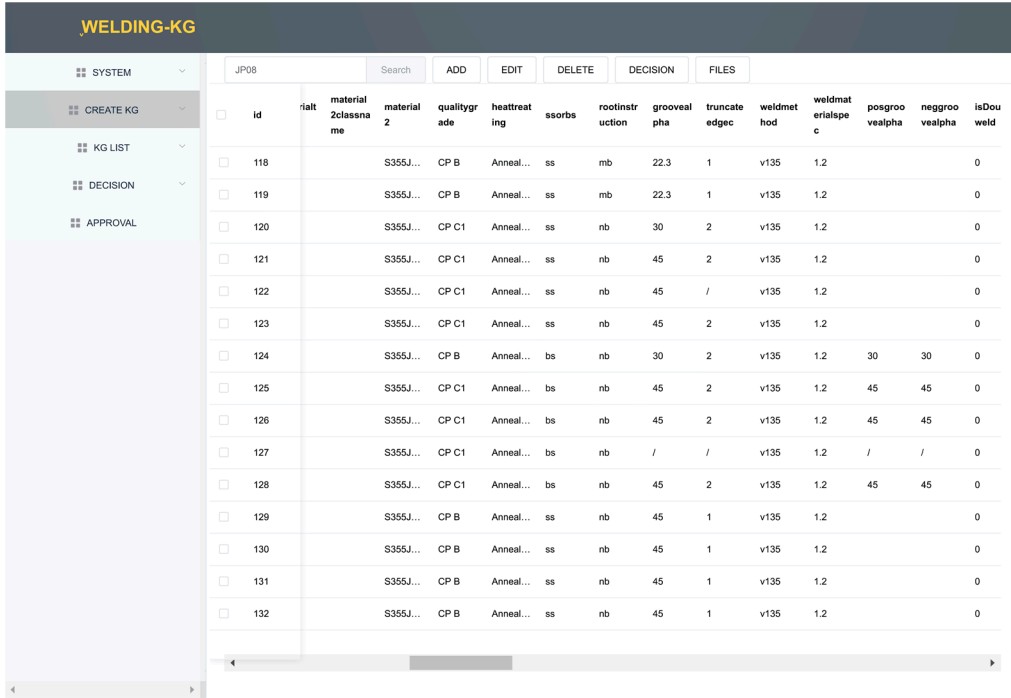

**Figure 5.** Batch data decision making.

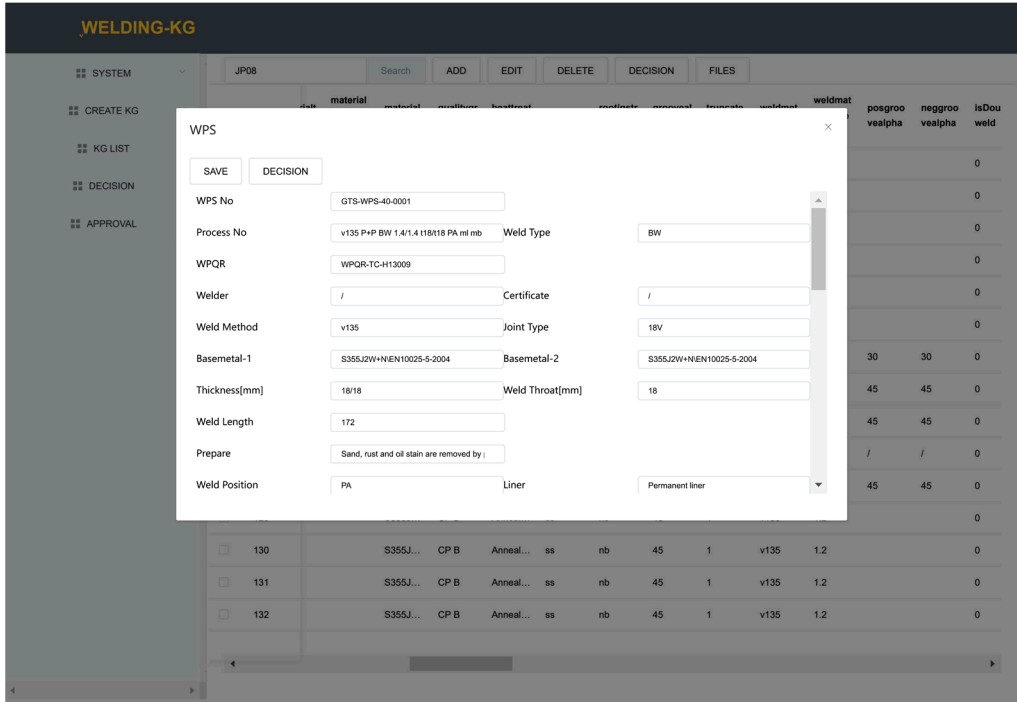

**Figure 6.** Structured document decision and view.

Figures 7 and 8 causal decision-related pages. As shown in Figure 1, the manually created and automatically extracted entities and relationships in the form of nodes and edges constitute the domain knowledge graph. We can obtain information about entities and relationships of interest by entering keywords and finding information about arrays by clicking on the view data button. Figure 8 shows the cause-and-effect guided decision process. Decision questions are entered in natural language, and decision results are presented in natural language and knowledge graphs. In addition, we can select the

question area on the left side of the search box to improve the efficiency and accuracy of the decision. This decision result can provide guiding suggestions for basic problem queries and complex engineering problem solving. For example, reference standards for bogie welding tasks are available, the terminology is quickly retrieved, and the traceability of welding quality is supported.

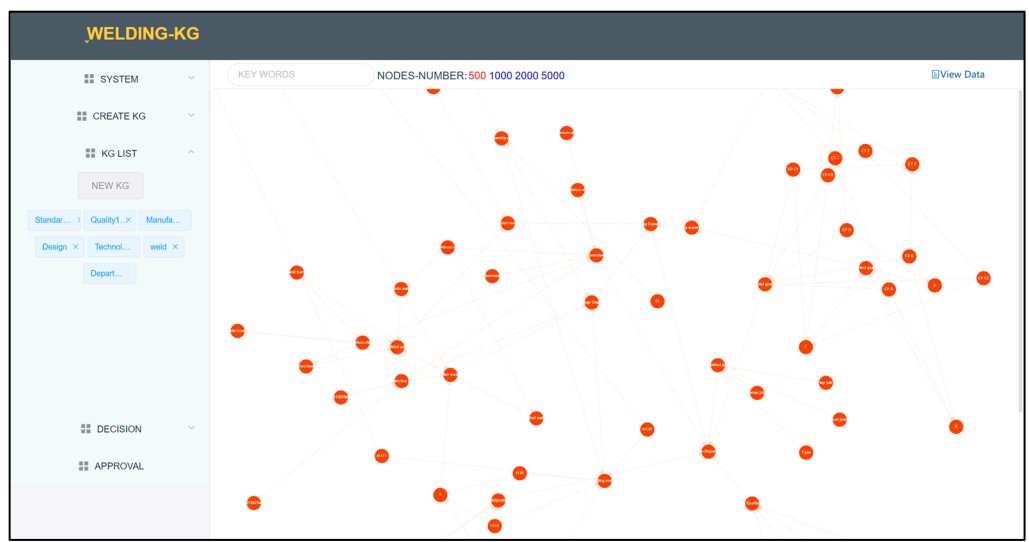

**Figure 7.** Knowledge graph view and search.

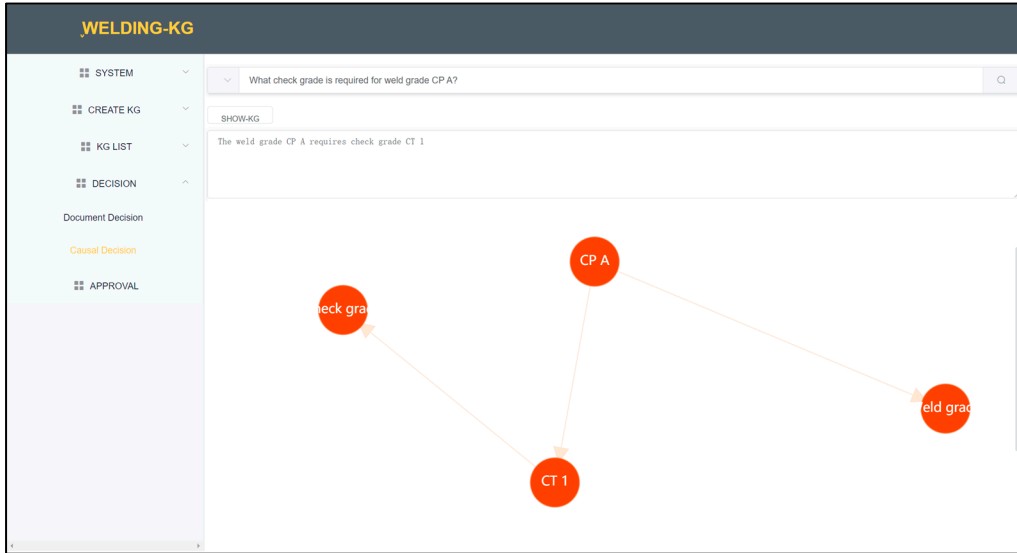

**Figure 8.** Causal decision process.

## 5. Conclusions

A hybrid model is designed that combines CBR, RBR, and knowledge graphs to overcome some limitations in welding decision making. A case retrieval algorithm based on weights and a knowledge graph oriented to the welding cycle were constructed. Entity identification, relationship extraction, and hybrid decision models were validated with engineering data. As a result, optimal entity (CRF, 0.710 for F1-score) and relational (BiLSTM + Attention, 0.816 for F1-score) models for our data were used for knowledge graph construction. The decision accuracy for structured documents is 0.947, and practical guidance is provided for causal issues.

The proposed hybrid decision-making approach is beneficial for intelligent decision making in welding engineering. Such an innovative means of introducing knowledge

graphs into welding decision systems enriches the domain of decision theory. In addition, this study may be helpful for decision making in other areas, such as casting and forging.

Although this study can support most decision problems, there are limitations to multi-entity and multi-relationship decisions. Therefore, optimizing the extraction, storage, and retrieval of multi-entity and multi-relationship decisions may become the focus of future research.

**Author Contributions:** K.G. and X.Y. proposed the idea of constructing a hybrid decision model and were the main contributors to writing the manuscript; Y.S. summarized the problems in engineering decision making and supported the description of the bogie manufacturing process; G.Y. provided technical support for the software architecture design and system construction; X.Y. developed a feasible experimental protocol and reviewed the manuscript. All authors have read and agreed to the published version of the manuscript.

**Funding:** This work was supported by the National Natural Science Foundation of China (Grant numbers: 51875072 and 52005071) and the Foundation for Overseas Talents Training Project in Liaoning Colleges and Universities (Grant number: 2018LNGXGJWPY-YB012).

**Institutional Review Board Statement:** Not applicable.

**Informed Consent Statement:** Not applicable.

**Data Availability Statement:** The data involved in the paper are available upon reasonable request to the corresponding author.

**Acknowledgments:** The authors are grateful for the support of the National Natural Science Foundation of China (Grant numbers: 51875072 and 52005071) and the Foundation for Overseas Talents Training Project in Liaoning Colleges and Universities (Grant number: 2018LNGXGJWPY-YB012).

**Conflicts of Interest:** The authors declare no conflict of interest.

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
