# Peer review of "Hybrid Decision-Making-Method-Based Intelligent System for Integrated Bogie Welding Manufacturing"

_asi, doi:10.3390/asi6010029_

Round 1

Reviewer 1 Report

The topic raised is quite interesting, but needs clarification and the addition of a few things:

1. The introduction must explain what has happened so far in bogie welding.

2. It is necessary to include the parameters in the bogie welding

3. Statement line 369 "The low case matching accuracy is mainly due to the incomplete case base, especially the lack of “Crossbeam” and “Brake hanger/beam” cases". Needs to be explained more comprehensively

4. Statement line 413 "This decision result can provide guiding suggestions for basic problem queries and complex engineering problem-solving". It is necessary to convey what recommendations are obtained on bogie welding

Author Response

Original Manuscript ID: asi-2159389

Original Article Title: "Hybrid decision-making method-based intelligent system for integrated bogie welding manufacturing"

To: Editor

Re: Response to reviewers

Dear Editor, Dear Reviewers,

Thank you very much for the insightful comments and for giving us a choice to correct the manuscript’s shortcoming. We already read the comments carefully and revised the manuscript according to the valuable suggestions at the first time. We hope that this revision will make the manuscript meet the publisher’s requirements.

The responses to the comments point by point are listed below. Please feel free to contact us with any questions. In addition, we have checked our language issues and cited appropriate references according to the comments received. If the revised manuscript may have shortcomings, please tell us. We will try our best to continue to better our manuscript to improve our manuscript. Really thank your insightful comments and help again!

Correspondence about this paper should be directed to X.H. Yang at the following address and e-mail.

Address:  School of Materials Science and Engineering, Dalian Jiaotong University, Dalian 116028, China.

e-mail:  yangxh@djtu.edu.cn

Thanks very much again for your attention to our paper. Once again, thank you for your help in processing our paper.

Yours sincerely,

Xinhua Yang

Best regards,

<K.N  Guan > et al.

Reviewers' comments to the authors: 

Reviewer#1, Concern # 1: The introduction must explain what has happened so far in bogie welding.

Author response:  Thank you for your comments. We have revised the introduction of the manuscript according to your comments.

Author action:  We have summarized and added the current status of the bogie manufacturing decision-making process in the introduction's first paragraph.

However, due to the high quality, standards, and engineering complexity of bogie welding, the manufacturing process is still dominated by manual decisions, supplemented only by intelligent decisions in single-stage manufacturing.

Reviewer#1, Concern # 2: It is necessary to include the parameters in the bogie welding.

Author response:  Thank you for your comments. We have revised the manuscript according to your comments.

Author action:  We have added the markers in Figure 1 and illustrated them in the description. We also refine the data in the validation section in the first paragraph of section 4.2.

Reviewer#1, Concern # 3: Statement line 369 "The low case matching accuracy is mainly due to the incomplete case base, especially the lack of “Crossbeam” and “Brake hanger/beam” cases". Needs to be explained more comprehensively.

Author response:  Thank you for your comments. We have added information based on your comments.

Author action:  We have added explanatory information in the second paragraph of section 4.2.

This is probably because, in actual production, there are fewer welding processes in this part compared to “Vehicle Frame” and “Side Beams,” thus leading to a lack of accumulated standardized engineering data.

Reviewer#1, Concern # 4: Statement line 413 "This decision result can provide guiding suggestions for basic problem queries and complex engineering problem-solving". It is necessary to convey what recommendations are obtained on bogie welding.

Author response:  Thank you for your comments. We have added explanatory information to the end of the sentence.

Author action:  We have revised the manuscript at the end of section 4.

This decision result can provide guiding suggestions for basic problem queries and complex engineering problem-solving. For example, reference standards for bogie welding tasks are available, the terminology is quickly retrieved, and traceability of welding quality is supported.

And so on, please read our revised manuscript. We thank the comments and the opportunity for us to improve our manuscript. As much as possible, the questions were taken into account during the preparation of the revised manuscript. We hope that the manuscript is now suitable for publication.

Reviewer 2 Report

The description of the topic is very detailed but without authentic scientific sound. The authors try to describe the problems with the coordination and inspection of the bogie welding system, not focusing on the welding process directly on the welding parameters. The paper is written at a very general level. Mentioned results are not based on scientific research.

I have several comments:

* Fig. 1: The description of the fig. is missing

* A lot of input of welding parameters are used, not focusing on the parameters that have a real impact on the joint quality.

* Abbreviations are not explained in sentences

* Figs. 5,6,7,8 are not readable in the printed version

*Line 140 - 141 - The same sentences

* How the authors verified the obtained results?

* How can the results be applied in welding process manufacturing?

* How can this knowledge help increase the welding process quality?

Author Response

Original Manuscript ID: asi-2159389

Original Article Title: "Hybrid decision-making method-based intelligent system for integrated bogie welding manufacturing"

To: Editor

Re: Response to reviewers

Dear Editor, Dear Reviewers,

Thank you very much for the insightful comments and for giving us a choice to correct the manuscript’s shortcoming. We already read the comments carefully and revised the manuscript according to the valuable suggestions at the first time. We hope that this revision will make the manuscript meet the publisher’s requirements.

The responses to the comments point by point are listed below. Please feel free to contact us with any questions. In addition, we have checked our language issues and cited appropriate references according to the comments received. If the revised manuscript may have shortcomings, please tell us. We will try our best to continue to better our manuscript to improve our manuscript. Really thank your insightful comments and help again!

Correspondence about this paper should be directed to X.H. Yang at the following address and e-mail.

Address:  School of Materials Science and Engineering, Dalian Jiaotong University, Dalian 116028, China.

e-mail:  yangxh@djtu.edu.cn

Thanks very much again for your attention to our paper. Once again, thank you for your help in processing our paper.

Yours sincerely,

Xinhua Yang

Best regards,

<K.N  Guan > et al.

Reviewers' comments to the authors: Reviewer#2, Concern # 1: Fig. 1: The description of the fig. is missing.

Author response:  Thank you for your comments. We have revised the description of Fig. 1.

Author action:  We have added the markers in Figure 1 and illustrated them in the description.

Figure 1. Description of bogie structure. 1- Framework components, 2- Series II transverse shock absorbers seats, 3- Crossbeam components, 4- Spring mounts, 5- Lifting seat, 6- Side beam components, 7- Air spring seats, 8- Axle box locator, 9- Series I pendant locators.

Reviewer#2, Concern # 2: A lot of input of welding parameters are used, not focusing on the parameters that have a real impact on the joint quality.

Author response:  Thank you for your comments. Your comments are constructive, and we will work on specific issues in the follow-up. However, our main objective in this work is to achieve decision support for the full bogie welding and manufacturing cycle, which is a necessary foundation for our subsequent research. Therefore, the parameters we have chosen may not be directly related to the quality of the weld but are necessary for the welding process.

Reviewer#2, Concern # 3: Abbreviations are not explained in sentences.

Author response:  Thank you for your comments. We have checked the manuscript and added explanations.

Author action:  We have added explanatory information in the first paragraph of section 2.3 and the third paragraph of section 3.3.

Reviewer#2, Concern # 4: Figs. 5,6,7,8 are not readable in the printed version.

Author response:  Thank you for your comments.

Author action:  We have increased the resolution of the figures. We also provide editable original documents.

Reviewer#2, Concern # 5: Line 140 - 141 - The same sentences.

Author response:  Thank you for your comments on this detailed question, we have removed the duplicate sentences.

Author action:  Removed sentence “In addition, each assignment phase corresponds to some specific professional content.”

Reviewer#2, Concern # 6: How the authors verified the obtained results?

Author response:  Thank you for your comments. We have validated the structured document decision task by collecting standardized process data from welding documents approved by experts and used in actual production. We verified the correctness of the cause-and-effect-based guided decision-making task using a question-and-answer format and compared the results with the standard.

Author action:  We have modified and added validation information to the first paragraph of section 4.2.

The 284 welding process data of a type of bogie were used for system testing, including 107 side beams, 55 cross beams, 55 frames, and 22 brake hangers and beams. The data is mainly derived from WPS data that has been approved by experts and applied to engineering production. Define design information as known data (type of base material, size of base material, quality grade, joint form, etc.) and process information (welding position, assembly gap, blunt edge, welding parameters, preheating temperature, etc.) as decision data. The model completes the validation of the decision information based on known information. The ratio of correct cases to the number of test cases, Case Accuracy, is used to evaluate the accuracy of case matching. The accuracy of corrected cases (Corrected Accuracy) is calculated by the ratio of correct cases (Case Accuracy) to test cases (Test Cases). The test results are shown in Table 4.

Reviewer#2, Concern # 7: How can the results be applied in welding process manufacturing?

Author response:  Thank you for your comments. This part of the research work was applied to the bogie welding manufacturing process in the form of an intelligent system. This mainly includes decision-making with structured documentation and guided decision-making based on cause and effect.

Author action:  We have added the application information in the first paragraph of section 4.3.

Hybrid decision-making methods are applied to the bogie welding process in the form of intelligent systems, which consist mainly of structured documented decisions and cause-and-effect-based guided decision-making tasks. Structured documented decisions are used for the rapid development of WPS, welding schedules, inspection schedules, etc. The information to be decided is obtained by pre-inputting known data and based on a decision inference system to guide the welding production. Cause-and-effect-based decision-making is mainly for unstructured data. For example, the rapid determination of relevant standards in joint design, the acquisition of recommendations for inspection levels and means in the quality inspection, and the provision of reference information for the traceability of welding defects.

Reviewer#2, Concern # 8: How can this knowledge help increase the welding process quality?

Author response:  Thank you for your comments. The correctness of knowledge and knowledge-based decisions is the key to the welding process quality. We build knowledge systems based on domain standards and production-validated documentation to ensure knowledge correctness. Digital-based decision-making methods reduce the arbitrariness and errors of manual decisions and indirectly improve the overall weld quality.

And so on, please read our revised manuscript. We thank the comments and the opportunity for us to improve our manuscript. As much as possible, the questions were taken into account during the preparation of the revised manuscript. We hope that the manuscript is now suitable for publication.

Reviewer 3 Report

Please read the attachment. Thank you. 

Author Response

Original Manuscript ID: asi-2159389

Original Article Title: "Hybrid decision-making method-based intelligent system for integrated bogie welding manufacturing"

To: Editor

Re: Response to reviewers

Dear Editor, Dear Reviewers,

Thank you very much for the insightful comments and for giving us a choice to correct the manuscript’s shortcoming. We already read the comments carefully and revised the manuscript according to the valuable suggestions at the first time. We hope that this revision will make the manuscript meet the publisher’s requirements.

The responses to the comments point by point are listed below. Please feel free to contact us with any questions. In addition, we have checked our language issues and cited appropriate references according to the comments received. If the revised manuscript may have shortcomings, please tell us. We will try our best to continue to better our manuscript to improve our manuscript. Really thank your insightful comments and help again!

Correspondence about this paper should be directed to X.H. Yang at the following address and e-mail.

Address:  School of Materials Science and Engineering, Dalian Jiaotong University, Dalian 116028, China.

e-mail:  yangxh@djtu.edu.cn

Thanks very much again for your attention to our paper. Once again, thank you for your help in processing our paper.

Yours sincerely,

Xinhua Yang

Best regards,

<K.N  Guan > et al.

Reviewers' comments to the authors: 

Reviewer#3, Concern # 1: The manuscript should be formatted as the journal template.

Author response:  Thank you for your comments.

Author action:  We have revised the manuscript according to the journal template.

Reviewer#3, Concern # 2: Keywords: Please lowercase the keywords. Please provide between 5 and 10 keywords that should not repeat the words/phrases that appeared in the manuscript title.

Author response:  Thank you for your comments on keywords.

Author action:  We have modified the keywords according to your comments.

Keywords: welding; case-based; rule-based; knowledge graph; entity classification; relationship extraction

Reviewer#3, Concern # 3: Please add an outline for the manuscript at the end of the first section (introduction).

Author response:  Thank you for your comments. We have revised the manuscript according to your comments.

Author action:  We have added the outline at the end of the introduction.

The rest of this paper is organized as follows. Section 2, the knowledge model and hybrid decision scheme are constructed for the bogie welding manufacturing process. Section 3 implements the hybrid decision scheme based on case matching, rule base, and knowledge graph construction. Section 4 gives the resultant metrics of the comparative models for entity identification and relationship extraction in the construction of domain knowledge graphs. In Section 5, experimental results on the knowledge graph construction process, structured data decision-making, and causal decision-making are provided and analyzed. In Section 6, the corresponding conclusions are given.

Reviewer#3, Concern # 4: Line 127: the reviewer suggests you enhance the resolution of figure 1.

Author response:  Thank you for your comments.

Author action:  We have re-edited Figure 1 and enhanced the resolution.

Reviewer#3, Concern # 5: Line 400, 403, 415, and 417: Figures 5 and 6 appear to have a shallow resolution.

Author response:  Thank you for your comments.

Author action:  We have increased the resolution of the figures.

Reviewer#3, Concern # 6: Literature review: for the welding approach. The author could refer to the following work https://doi.org/10.3390/mi13111890. Please add the other MCDM approaches that should have been mentioned in your introduction. The hybrid method of Malmquist and DEA in this study https://doi.org/10.3390/drones6110363. For the other assessment, a framework based on muli-criteria decision making (MCDM) is provided that integrates spherical fuzzy Analytical Hierarchical Process (SF-AHP) and grey Complex Proportional Assessment (G-COPRAS), in which spherical fuzzy sets and grey numbers are used to express the ambiguous linguistic evaluation statements of experts as presented in https://doi.org/10.3390/axioms11050228.

Author response:  Thank you for your comments. We have carefully read the relevant papers and cited relevant work.

Author action:  We have added references relevant to our paper in the second paragraph of the introduction.

Wang, C.-N.; Yang, F.-C.; Vo, N.T.M.; Nguyen, V.T.T. Wireless Communications for Data Security: Efficiency Assessment of Cybersecurity Industry—A Promising Application for UAVs. Drones 2022, 6, 363.

Dang, T.-T.; Nguyen, N.-A.-T.; Nguyen, V.-T.-T.; Dang, L.-T.-H. A Two-Stage Multi-Criteria Supplier Selection Model for Sustainable Automotive Supply Chain under Uncertainty. Axioms 2022, 11, 228.

Reviewer#3, Concern # 7: All equations should be marked with a number and mentioned or explained in the text. References: The authors do not have sufficient concerns in this paper. Standard research should have at least 35-65 references; in this paper, you have only 20 references. All references should be formatted as the guide of the journal template.

Author response:  Thank you for your comments. We have further summarized the relevant studies.

Author action:  We have added relevant research to the introduction and cited the referenced papers.

Reviewer#3, Concern # 8: In Section 5. Conclusions: it is suggested to summarize the significance of your study process. Besides, the authors should emphasize their contributions and findings to this study. Similarly, please present the study limitation and their further studies.

Author response:  Thank you for your comments. We have revised the conclusion of the manuscript according to your comments.

Author action:  We have revised and subdivided the conclusion to make it clearer.

A hybrid model is designed in collaboration with CBR, RBR, and knowledge graph to solve some limitations in welding decision-making. The case retrieval algorithm based on weight and the knowledge graph oriented to the welding cycle were constructed. Entity identification, relationship extraction, and hybrid decision models were validated with engineering data. As a result, optimal entity (CRF, 0.710 for F1-score) and relational (BiLSTM+Attention, 0.816 for F1-score) models for our data are used for knowledge graph construction. The decision accuracy for structured documents is 0.947, and practical guidance is provided for causal issues.

The proposed hybrid decision-making approach is beneficial for intelligent decision-making in welding engineering. An innovative means of introducing knowledge graphs into welding decision systems enrich the domain of decision theory. In addition, this study may be helpful for decision-making in other areas, such as casting and forging.

Although this study can support most decision problems, there are limitations to multi-entity and multi-relationship decisions. Therefore, optimizing extraction, storage, and retrieval of multi-entity and multi-relationship may become the following research focus.

Reviewer#3, Concern # 9: How did you validate the results?

Author response:  We have validated the structured document decision task by collecting standardized process data from welding documents approved by experts and used in actual production. We verified the correctness of the cause-and-effect-based guided decision-making task using a question-and-answer format and compared the results with the standard.

Author action:  We have modified and added validation information to the first paragraph of section 4.2.

The 284 welding process data of a type of bogie were used for system testing, including 107 side beams, 55 cross beams, 55 frames, and 22 brake hangers and beams. The data is mainly derived from WPS data that has been approved by experts and applied to engineering production. Define design information as known data (type of base material, size of base material, quality grade, joint form, etc.) and process information (welding position, assembly gap, blunt edge, welding parameters, preheating temperature, etc.) as decision data. The model completes the validation of the decision information based on known information. The ratio of correct cases to the number of test cases, Case Accuracy, is used to evaluate the accuracy of case matching. The accuracy of corrected cases (Corrected Accuracy) is calculated by the ratio of correct cases (Case Accuracy) to test cases (Test Cases). The test results are shown in Table 4.

Reviewer#3, Concern # 10: What are the main limitations of this approach?

Author response:  Thank you for your comments. In cause-oriented decision-making, the knowledge of bogie welding is characterized by multiple entities and relationships due to its complex engineering characteristics. This study only focuses on extracting two-entity and single-relationships, so multi-entity and multi-relationship extraction are limited.

Author action:  We have described the study limitations in the third paragraph of our conclusions.

Although this study can support most decision problems, there are limitations to multi-entity and multi-relationship decisions. Therefore, optimizing extraction, storage, and retrieval of multi-entity and multi-relationship may become the following research focus.

Reviewer#3, Concern # 11: Have the authors performed the experimental study in this study? How did you evaluate the accuracy of this research? And what is the limitation of this study?

Author response:  Our experiments focus on entity identification and relationship extraction in knowledge graph construction and structured and cause-effect-based guided decision-making in decision making.

We validate entity and relationship models through collected test datasets and verify decision-making tasks through actual valid engineering data. We have added validation information in the first paragraph of section 4.2.

The work is mainly restricted to constructing multi-entity and multi-relationship-based knowledge, which we also describe in the conclusion.

And so on, please read our revised manuscript. We thank the comments and the opportunity for us to improve our manuscript. As much as possible, the questions were taken into account during the preparation of the revised manuscript. We hope that the manuscript is now suitable for publication.

Round 2

Reviewer 2 Report

All revisions were done, just fig. 5. and 6. are still not readable.

Author Response

Dear Reviewer,

We feel great thanks for your professional review work on our article. We have revised Figures 5 and 6 based on your comments. Thanks very much again for your attention to our paper. Once again, thank you for your help in processing our paper.

Correspondence about this paper should be directed to X.H. Yang at the following address and e-mail.

Address:  School of Materials Science and Engineering, Dalian Jiaotong University, Dalian 116028, China.

e-mail:  yangxh@djtu.edu.cn

Yours sincerely,

Xinhua Yang

Best regards,

<K.N  Guan > et al.

Reviewer 3 Report

Dear Editor and Authors:

Thank you for providing the point-to-point response.

The authors have carefully and patiently corrected and answered the comments and quetions. The manuscript sounds perfect now. The reviewer strongly suggests it be accepted for publication.

Thank you for reading.

Sincerely yours,

The reviewer.

Author Response

To: Editor

Re: Response to reviewers

Dear Editor, Dear Reviewers,

We feel great thanks for your professional review work on our article. We have carefully checked the manuscript and made some changes to spelling and grammatical errors in the manuscript. These changes do not affect the content or the framework of the thesis. Really thank your insightful comments and help again!

Correspondence about this paper should be directed to X.H. Yang at the following address and e-mail.

Address:  School of Materials Science and Engineering, Dalian Jiaotong University, Dalian 116028, China.

e-mail:  yangxh@djtu.edu.cn

Thanks very much again for your attention to our paper. Once again, thank you for your help in processing our paper.

Yours sincerely,

Xinhua Yang

Best regards,

<K.N  Guan > et al.